# Flavonoids and Phenols, the Potential Anti-Diabetic Compounds from *Bauhinia strychnifolia* Craib. Stem.

**DOI:** 10.3390/molecules27082393

**Published:** 2022-04-07

**Authors:** Rachanida Praparatana, Pattaravan Maliyam, Louis R. Barrows, Panupong Puttarak

**Affiliations:** 1Department of Pharmacognosy and Pharmaceutical Botany, Faculty of Pharmaceutical Sciences, Prince of Songkla University, Hat-Yai, Songkhla 90112, Thailand; rachanida.pra@gmail.com (R.P.); 5910720014@email.psu.ac.th (P.M.); 2Department of Pharmacology and Toxicology, University of Utah, Salt Lake City, UT 81112, USA; lbarrows@pharm.utah.edu; 3Phytomedicine and Pharmaceutical Biotechnology Excellence Center, Faculty of Pharmaceutical Sciences, Prince of Songkla University, Hat-Yai, Songkhla 90112, Thailand

**Keywords:** diabetes, *Bauhinia strychnifolia*, α-glucosidase, dipeptidyl peptidase-IV, glucose uptake, flavonoids

## Abstract

Bioactive compounds from medicinal plants are good alternative treatments for T2DM. They are also sources of lead molecules that could lead to new drug discoveries. In this study, *Bauhinia strychnifolia* Craib. stem, a traditional Thai medicinal plant for detoxification, was extracted into five fractions, including crude extract, BsH, BsD, BsE, and BsW, by ethanolic maceration and sequential partition with hexane, dichloromethane, ethyl acetate, and water, respectively. Among these fractions, BsE contained the highest amounts of phenolics (620.67 mg GAE/g extract) and flavonoids (131.35 mg QE/g extract). BsE exhibited the maximum inhibitory activity against α-glucosidase (IC_50_ 1.51 ± 0.01 µg/mL) and DPP-IV (IC_50_ 2.62 ± 0.03 µg/mL), as well as dominantly promoting glucose uptake on 3T3-L1 adipocytes. Furthermore, the four compounds isolated from the BsE fraction, namely resveratrol, epicatechin, quercetin, and gallic acid, were identified. Quercetin demonstrated the highest inhibitory capacity against α-glucosidase (IC_50_ 6.26 ± 0.36 µM) and DPP-IV (IC_50_ 8.25 µM). In addition, quercetin prominently enhanced the glucose uptake stimulation effect on 3T3-L1 adipocytes. Altogether, we concluded that quercetin was probably the principal bioactive compound of the *B**. strychnifolia* stem for anti-diabetic, and the flavonoid-rich fraction may be sufficiently potent to be an alternative treatment for blood sugar control.

## 1. Introduction

Diabetes mellitus (DM) is a chronic metabolic disorder characterized by abnormally high levels of blood glucose [1]. Over 460 million people worldwide have been affected by DM, and this number has been projected to increase continuously [2,3]. From all diabetes cases, type 2 diabetes mellitus (T2DM) is the most common type, accounting for around 90–95% [4]. It is primarily caused by insulin-resistance, lack of secretion, and/or inefficient action of insulin hormone, contributing to hyperglycemia [2]. T2DM is one of the serious health problems in many countries, including Thailand, and leads to reducing life quality and increasing mortality and healthcare costs [5]. Glycemic control is the critical step of T2DM management and is very important for the prevention of long-term hyperglycemia complications, such as cardiovascular disease, neuropathy, nephropathy, retinopathy, foot damage, and hearing impairment [6,7].

Recently, many anti-diabetic drugs have been developed and consist of multiple modes of action to control blood sugar levels, such as inhibiting metabolic enzymes (such as α-amylase and α-glucosidase), blocking the dipeptidyl peptidase-IV (DPP-IV) enzyme, and enhancing glucose uptake [8,9]. However, there is no successful remedy for curing T2DM. Undesirable side effects, such as diarrhea, abdominal distention, flatulence, liver disorder, bloating, and nausea, are key and cause a lack of patient compliance and poor remedy effectiveness [10,11]. Therefore, new glucose-lowering agents from natural plants that might express less, or no, side effects would be beneficial.

*Bauhinia strychnifolia* Craib., a local Thai herb, commonly known as Yanang Dang, is one of the candidate plants for T2DM treatment. *B. strychnifolia* is a reddish climbing plant, mostly found in the north of Thailand. *B. strychnifolia* contains high phenolic and flavonoid compounds, which are considered likely bioactive compounds for various biological activities, including anti-diabetic [12,13,14]. Formerly, the leaves, roots, and stems of *B**. strychnifolia* have all been used in foods, health tea drinks, tonics, and traditional Thai medicines (TTM) for immunization, nourishment, alleviating fever, reducing alcoholic toxification, allergy treatment, eliminating toxins, anti-inflammation, and anti-cancer and -diarrheal effects [14,15,16,17]. Several studies have also reported its biological and pharmacological activities, which are consistent with TTM knowledge, such as antipyretic, alleviating allergy, detoxification, antioxidants, antimicrobial, anti-cancer, anti-HIV, and anti-diabetic [14,15,16,17,18,19]. Considering anti-diabetic activity, only two studies have been reported. It was found that the ethanolic extract of the *B**. strychnifolia* stem inhibited α-glucosidase in vitro and reduced glucose levels, triglycerides, and total cholesterol in the circulation of the alloxan-diabetic rats [18,19]. In addition, the two flavonoids, namely 3,5,7,3′,5′-pentahydroxy-flavanonol-3-O-a-L-rhamnopyranoside and 3,5,7-trihydroxychromone-3-O-a-L-rhamnopyranoside, that were isolated from the ethanolic extract of the *B**. strychnifolia* stem also exhibited good inhibitory effect against α-glucosidase [18]. However, other glucose-lowering effects of *B. strychnifolia* stem have not been studied, with its major bioactive compounds still unknown. Herein, we determined the different anti-diabetic effects of the *B. strychnifolia* stem and explored its potent bioactive compounds. We expected that these findings could support the use of *B. strychnifolia* as an alternative treatment for T2DM or source of lead molecules for new drug discovery in the future.

## 2. Results and Discussions

### 2.1. B. strychnifolia Stem Extraction

Mimicking a traditional method, the dried powder of B. strychnifolia stem was extracted by ethanolic maceration to obtain crude extract. Traditionally, ethanolic crude extract was used for the treatment of various ailments, such as fever, diarrhea, cancer, and infection [14,15,16,17]. In this study, we used crude extract that has a reddish-brown appearance as a main extract for screening anti-diabetic activities. Subsequently, the crude extract was further separated into four fractions, including BsH, BsD, BsE, and BsW, according to its polarity by using hexane, dichloromethane, ethyl acetate, and water, respectively. Four partitioned fractions were then investigated for anti-diabetic effects, and the most active fraction was selected for the isolation of bioactive compounds.

Extraction is a crucial step for recovering phytochemicals from plant materials. Its efficiency depends on the solvent polarity, chemical nature of the compounds, temperature, interfering substance, times, and skill of the extractor [20]. From Table 1, the results showed that the yield of extraction ranged from 0.40% to 60%. The BsW fraction gave the highest yield (60%) by weight, followed by crude extract (10.30%), BsD (8.15%), BsE (5.90%), and BsH (0.40%) fractions, respectively. This suggested that the increasing polarity of solvent will increase extraction yield

### 2.2. Total Phenolic Contents (TPC) and Total Flavonoid Contents (TFC)

Flavonoids are a large group of natural substances, containing variable phenolic structures, and mostly found in fruits, vegetables, nuts, tea, and herbs [21,22]. Previous studies demonstrated that most dietary flavonoids provided various medical activities, including anti-diabetic agents [21,22,23]. For example, rutin can reduce carbohydrate absorption by inhibiting α-glucosidase enzyme, kaempferol can enhance glucose uptake, and luteolin can inhibit lipid synthesis [21,22,23]. Therefore, in this study, we mainly focused on the flavonoids that were suggested as bioactive compounds. To preliminary screen for the bioactive compounds in each fraction, the TPC and TFC of crude extract and four partitioned fractions were evaluated using colorimetric assays. TPC value was obtained from gallic acid calibration curve y = 0.0002x + 0.0492, R^2^ = 0.999 and expressed as the milligram gallic acid equivalent per gram extract (mg GAE/g extract), whereas TFC value was obtained from quercetin calibration curve y = 0.0008x + 0.0392, R^2^ = 1, and expressed as milligram quercetin equivalent per gram extract (mg QE/g extract). From Table 1, the results showed that the highest amounts of both TPC and TFC were found in BsE, with less in crude extract, BsW, BsD, and trace in BsH. Phenolics and flavonoids in medicinal plants and foods were the principal constituents responsible for various anti-diabetic activities [22]. The higher amounts of TPC and TFC are the indication of possible therapeutic activities of plant extracts [24,25]. Thus, BsE, which has the higher amounts of both TPC and TFC, was hypothesized to exhibit greater anti-diabetic activities than other fractions. For this reason, BsE was used as a representative fraction for the further isolation of bioactive compounds (Figure 1).

### 2.3. Liquid Chromatography Tandem Mass Spectrometry (LC-MS/MS) Analysis

Liquid chromatograph–quadrupole time-of-flight mass spectrometer (LC-QTOF/MS) was used to identify flavonoid and phenolic components in BsE fraction. Based on the retention time (RT) comparison, mass spectrometric obtained under both negative and positive electron spray ionization modes (ESI^−^/ESI^+^), mass error, data identification score, 25 flavonoids, and 8 phenolics were preliminarily identified. The flavonoids in BsE fraction have been reported as anti-diabetic agents, such as quercetin, kaempferol, naringenin, apigenin, and luteolin (Table 2) [26,27,28,29,30,31,32,33,34,35,36,37,38]. A recent systemic review showed that quercetin can reduce the serum glucose level, at the dose of 10 mg/kg of mice body weight [26]. Quercetin from berry extract was shown to induce glucose uptake via an insulin-independent 5′ adenosine monophosphate-activated protein kinase (AMPK) pathway [27]. In vitro studies revealed that kaempferol can enhance insulin secretion and glucose uptake through protein kinase C [28,29]. These results support the idea that BsE fraction contains numerous anti-diabetic flavonoids, and this (flavonoid-rich fraction) might exhibit good anti-hyperglycemia activities.

Furthermore, we found that the compounds isolated from the BsE fraction (Section 2.4), which included resveratrol, epicatechin, quercetin, and gallic acid, are consistent with our hypothesis (Figure 2B, Table 2).

### 2.4. Isolation

Based on bioassay guide isolation, four compounds were isolated from the BsE fraction using silica gel and the Sephadex LH20 column. Their chemical structures were identified as resveratrol, epicatechin, quercetin, and gallic acid. The ^1^H and ^13^C nuclear magnetic resonance (NMR) spectra of each compound, shown below, also correspond with the structure of resveratrol, epicatechin, quercetin, and gallic acid in databases. Thus, it was confirmed that chemical structures in Figure 1 were resveratrol, epicatechin, quercetin, and gallic acid. To date, only two flavonoids, namely 3,5,7-Trihydroxychromone-3-O-α-L-rhamnopyranoside and 3,5,7,3′,5′-pentahydroxy-flavanonol-3-O-α-L-rhamnopyranoside were isolated from *B. strychnifolia* stem [18]. Additionally reported were β-sitosterol and stigmasterol. This is the first report that identified these four compounds from the *B. strychnifolia* stem. All isolated compounds were further investigated for anti-diabetic activities.

#### 2.4.1. Resveratrol

^1^H-NMR (500 MHz, CD_3_OD): δ 7.35 (2H, d, J = 8.5 Hz, H2′, H6′), 6.96 (1H, d, J = 16.0 Hz, H8), 6.80 (3H, d, J = 16.0 Hz, H7), 6.44 (2H, d, J = 2.5 Hz, H2, H6), and 6.15 (1H, t, J = 2.0 Hz, H4); ^13^C-NMR (500 MHz, CD_3_OD): δ 141.3 (C1), 105.8 (C2, C6), 159.7 (C3, C5), 102.7 (C4), 129.4 (C7), 127.1 (C8), 130.5 (C1′), 128.8 (C2′, C6′), 116.5 (C3′, C5′), and 158.4 (C4′).

#### 2.4.2. Epicatechin

^1^H-NMR (500 MHz, CD_3_OD): 4.84 (br, s, H2), 4.18 (1H, m, H3), 2.87 (1H, dd, J = 5, 17.0 Hz, H4a), 2.74 (1H,dd, J = 3, 17 Hz, H4b), 5.91 (1H, d, J = 2.5 Hz, H6), 5.94 (1H, d, J = 2 Hz, H8), 6.97 (1H, J = 2 Hz, H2′), 6.76 (1H, d, J = 8 Hz, H5′), and 6.8 (1H, dd, J = 2,8.5 Hz, H6′); ^13^C-NMR (500 MHz, CD_3_OD): δ 79.8 (C-2), 67.5 (C-3), 29.2 (C-4), 157.3 (C-5), 96.4 (C-6), 157.9 (C-7), 95.9 (C-8), 157.6 (C-9), 100.1 (C-10), 132.3 (C-1′), 115.3 (C-2′), 145.9 (C-3′), 145.8 (C-4′), 115.9 (C-5′), and 119.4 (C-6′).

#### 2.4.3. Quercetin

^1^H-NMR (400 MHz, CD_3_OD): 12.48 (1H, s, 5-OH), 10.75 (1H, s, br, 3-OH), 9.56 (1H, s, br, 7-OH), 9.32 (2H, s, br, 2×-OH), 6.19 (1H, d, J = 2.0 Hz, H-6), 6.41 (1H, d, J = 2.0 Hz, H-8), 7.68 (1H, d, J = 2.2 Hz, H-2′), 6.89 (1H, d, J = 8.5 Hz, H-5′), 7.54 (1H, dd, J = 2.2, and 8.5 Hz, H-6′); ^13^C-NMR (100 MHz, CD_3_OD): δ 147.6 (s, C-2), 135.6 (s, C-3), 175.8 (s, C-4), 160.6 (s, C-5), 98.1 (d, C-6), 163.8 (s, C-7), 93.3 (d, C-8), 156.1 (s, C-9), 103.0 (s, C-10), 121.9 (s, C1), 115.0 (d, C-2), 145.0 (s, C-3), 146.7 (s, C-4), 115.5 (d, C-5), and 120.0 (d, C-6).

#### 2.4.4. Gallic Acid

^1^H-NMR (500MHz, CD_3_OD): δ 7.06 (2H, s, H-galloy l-2, 6); ^13^C-NMR (125MHz, CD_3_OD): 120.5 (C-1), 108.9 (C-2, 6), 145.0 (C-3, 5), 138.1 (C-4), and 168.9 (C-7).

### 2.5. α-Glucosidase Inhibitory Assay

In this study, we investigated the anti-diabetic effect of crude extract, partitioned fractions, and isolated compounds on α-glucosidase activity. As shown in Table 3, the results displayed that BsE fraction has the highest inhibitory activity, with an IC_50_ value of 1.51 ± 0.01 µg/mL, followed by crude extract (IC_50_ 2.37 ± 0.13 µg/mL), BsW (IC_50_ 2.37 ± 0.13 µg/mL), BsD (IC_50_ 10.09 ± 0.75 µg/mL), and BsH fractions (IC_50_ 10.61 ± 0.93 µg/mL), respectively. The inhibitory activity against α-glucosidase of BsE was significantly higher than acarbose, the positive control, which inhibited the enzyme with IC_50_ 329.48 ± 6.91 µg/mL. Moreover, four compounds isolated from BsE, namely resveratrol, epicatechin, quercetin, and gallic acid, were further investigated. Results demonstrated that quercetin has the highest inhibitory ability, with IC_50_ 6.26 ± 0.36 µM, followed by resveratrol, with IC_50_ 8.16 ± 10 µM, whereas epicatechin and gallic acid have no effect. IC_50_ values of both quercetin and resveratrol were also significantly lower than that of acarbose. Currently, acarbose, voglibose, and miglitol are synthetic drugs that serve as α-glucosidase inhibitors and are often prescribed for T2DM patients [39]. These drugs result in slowing down the digestion of polysaccharides from dietary into monosaccharides, followed by delaying glucose absorption and reducing postprandial hyperglycemia [40,41]. Thus, quercetin and resveratrol might be more potent than acarbose in reducing postprandial hyperglycemia. Moreover, as compared to the α-glucosidase inhibitory effect of known active flavonoids from the *B. strychnifolia* stem, namely 3,5,7-Trihydroxychromone-3-O-α-L-rhamnopyranoside (IC_50_ 540 µg/mL) and 3,5,7,3′,5′-pentahydroxy-flavanonol-3-O-α-L-rhamnopyranoside (IC_50_ 980 µg/mL) [18], it was found that quercetin and resveratrol were more potent. Overall, quercetin and resveratrol potentially inhibited α-glucosidase enzyme, showing a potential to decrease postprandial hyperglycemia.

### 2.6. Dipeptidyl Peptidase-IV (DPP-IV) Inhibitory Assay

DPP-IV is a serine protease enzyme produced from the intestine and associated with degradation of the incretin hormone (a stimulator of insulin secretion). Inhibiting DPP-IV leads to the increasing of active incretin hormone, followed by elevating insulin secretion [42,43,44]. Insulin regulates glucose homeostasis by inducing glucose storage in muscles, liver, and adipose tissues, stimulating triglyceride synthesis, as well as suppressing the release of free fatty acid into circulation [45]. Blocking the DPP-IV enzyme was another crucial strategy that reflected the glucose-lowering properties of crude extract, partitioned fractions, and isolated compounds. As demonstrated in Table 3, the results showed that BsE gave the highest activity, with an IC_50_ value of 2.62 ± 0.03 µg/mL, followed by BsW (IC_50_ 3.20 ± 0.02 µg/mL). Crude extract, BsD, and BsH exhibited an IC_50_ value higher than 50 µg/mL. BsE fraction showed good inhibitory activity, with low IC_50_, which close to positive control, diprotin A (IC_50_ 1.14 ± 0.05 µg/mL). Then, four compounds, isolated from BsE, were further investigated. The results showed that quercetin possessed the maximum ability, with an IC_50_ value of 2.49 ± 0.01 µg/mL (8.25 µM), followed by resveratrol (IC_50_ 3.22 ± 0.02 µg/mL or 14.11 µM), while epicatechin and gallic acid inhibited DPP-IV activity, with an IC_50_ value higher than 50 µM. The potency of quercetin, the best compound, was similar to diprotin A. Diprotin A is a standard DPP-IV inhibitor that function as anti-diabetic drugs, such as sitagliptin, vildagliptin, and saxagliptin [42]. Although no prior studies tested the DPP-IV inhibitory activity in extracts or compounds from the *B. strychnifolia* stem, it has been reported that synthetic resveratrol and commercial quercetin could inhibit the DPP-IV enzyme; based on molecular docking, both gave high docking scores, which indicates that they could bind to the active site of DPP-IV [46,47]. These data support that quercetin and resveratrol were DPP-IV inhibitors, even if their potencies were less than anti-diabetic drugs.

### 2.7. Cell Viability

We used 3T3-L1 adipocytes and raw 264.7 macrophage cells as models for further examining of the cellular mechanisms of crude extract, partitioned fractions, and isolated compounds. Cytotoxicity of all samples toward 3T3-L1 adipocytes and raw 264.7 cells were assessed using MTT assay. As shown in Figure 3, the results showed that the cell viability of both cells was more than 80% at every tested concentration of all samples, indicating no significant cytotoxicity of crude extract, partitioned fractions, and isolated compounds at any tested concentration. Thus, these results, as well as the historical use of *B. strychnifolia*, suggest that the crude extract, partitioned fractions, and isolated compounds are safe and could be used for further experiments.

### 2.8. Glucose Uptake in 3T3-L1 Adipocytes

To further evaluate the anti-diabetic effect of crude extract, partitioned fractions, and isolated compounds on cellular glucose uptake, we used differentiated 3T3-L1 adipocytes as a model, since adipocytes is one of the major issues involved in insulin function [45]. As demonstrated in Figure 4A, the results showed that crude extract, BsH, BsD, BsE, and BsW fractions could enhance glucose uptake in adult 3T3-L1 adipocytes in a different pattern. The BsE fraction displayed the highest percentage of glucose uptake stimulation at the lowest concentration (85.63% stimulation at 12.5 µg/mL), which is equivalent to positive controls, insulin (92.81% stimulation), and metformin (85.34% stimulation). Four compounds, isolated from BsE, were then investigated. The results showed that quercetin demonstrated a higher glucose uptake stimulation effect than resveratrol, gallic acid, and epicatechin, and its activity was also equivalent to positive controls, insulin (92.81% stimulation), and metformin (85.34% stimulation). Unlike most anti-diabetic drugs, metformin is derived from natural plants and used as a first-line drug for T2DM management [48]. It enhances cellular glucose uptake on muscle, liver, and adipocytes, mainly by activating the AMP-activated protein kinase (AMPK) [48]. This is the first preliminary study of extracts and their bioactive compounds from the *B. strychnifolia* stem on cellular glucose uptake. It has been reported that quercetin and 3-O-Acyl-epicatechin could promote glucose by increasing GLUT-4 translocation in skeletal muscle cells, while gallic acid also enhances glucose uptake by inducing GLUT-4 translocation in 3T3-L1 adipocytes [49,50,51]. At low dose, resveratrol increased glucose uptake and lipid accumulation in 3T3-L1 adipocytes via the insulin signaling pathway [52]. Altogether, the literature suggests that quercetin, resveratrol, epicatechin, and gallic acid from BsE fraction might promote glucose uptake in adipocytes via increasing GLUT-4 translocation.

### 2.9. Nitric Oxide (NO) Production

Previous studies reported that the excess nitric oxide (NO), an inflammatory cytokine, can cause insulin resistance in obesity [53]. Lipopolysaccharide (LPS)-induced raw 264.7 cells were used for investigation of the inhibitory effect of crude extract, partitioned fractions, and isolated compounds on nitric oxide production. As demonstrated in Figure 5, the results showed that BsE slightly inhibited NO-production of LPS-induced raw 264.7 cells, crude extract, and BsW slightly increased the production of NO, whereas BsD and BsH reduced NO production in a dose-dependent manner. Indomethacin, the positive control, showed potent anti-inflammatory, which was significantly higher than all fractions. Resveratrol, epicatechin, and gallic acid displayed an inhibitory effect on NO production at only high concentrations, but quercetin has no effect. These results indicated that the isolated compounds could inhibit nitric oxide production moderately. Therefore, it is possible that the low inhibitory effect of these isolated on NO production might be beneficial to improving insulin sensitivity, according to a mentioned study [54].

## 3. Materials and Methods

### 3.1. Chemicals and Reagents

The α-glucosidase from *Saccharomyces cerevisiae*, para-nitrophenyl-α-D-glucopyranoside (pNPG), acarbose, dipeptidyl peptidase-IV (DPP-IV) from porcine kidney, Gly-Pro-p-nitroanilide (GP-p-NA), diprotin A, lipopolysaccharide (LPS) from Escherichia coli O111: B4, dimethyl sulfoxide (DMSO), indomethacin, quercetin, gallic acid, human insulin, dexamethasone (DEX), isobutyl-methylxanthine (IBMX), metformin, and glucose (GO) assay kits were purchased from Sigma Aldrich Inc. (St. Louis, MO. USA). Folin–Ciocalteu reagent was bought from Merck (Darmstadt, Germany). Fetal bovine serum (FBS), penicillin-streptomycin, Dulbecco’s Modified Eagle Medium, high glucose (DMEM, high glucose), Roswell Park Memorial Institute (RPMI) 1640 medium, trypsin-EDTA, trypan blue dyes, and 3-(4,5-dimethylthiazol-2-yl)-2,5 diphenyltetrazolium bromide (MTT) were obtained from Thermo Fisher Scientific (Waltham, CA, USA). Ethanol, methanol, hexane, dichloromethane, ethyl acetate, and acetone (as analytical grades) were purchased from Labscan Asia Co., Bangkok, Thailand.

### 3.2. Plant Materials

*B. strychnifolia* stems were collected from the medical herbal garden at Prince of Songkla University, Thailand, in 2021. The herbarium voucher specimen was deposited in the Department of Biology, Faculty of Sciences, Prince of Songkhla University, Thailand, under the code number 0015181. Initially, plant stems were cleaned with tap water, cut into small segments, and dried in a hot air oven at 60 °C. After two days, the dried plant was ground into a fine powder and kept at RT until used.

### 3.3. Preparation of B. strychnifolia Extracts and Its Isolated Compounds

A total of 900 g of *B. strychnifolia* powder were macerated with 95% ethanol at RT for three days in triplicates. Subsequently, pooled ethanolic macerates were filtrated through filter papers, evaporated with a rotary evaporator at 50 °C, and freeze-dried to get crude extract. After that, 80 g of aliquoted crude extract were dissolved in 10% methanol and sequentially partitioned with hexane, dichloromethane, ethyl acetate, and water, respectively. Each partitioned solvent was then filtrated, evaporated, and freeze-dried to obtain hexane (BsH), dichloromethane (BsD), ethyl acetate (BsE), and water (BsW) fractions.

The BsE fraction that contained the highest amounts of flavonoids and exhibited good activity in enzyme assays was selected for further isolation of bioactive compounds. BsE was applied into silica gel column and eluted by a gradient system, starting with hexane in ethyl acetate, followed by mixtures of ethyl acetate and methanol, as well as mixtures of methanol and acetone. Obtained fractions were screened by TLC and re-chromatographed in silica gel with various gradient elution. Then, all compounds were re-purified by Sephadex LH20 and structurally identified by NMR analysis.

Crude extract, partitioned fractions, and isolated pure compounds were stored at 4 °C in the dark for further analysis.

### 3.4. Total Phenolic Contents (TPC)

In order to determine the total phenolic contents in crude extract and four partitioned fractions, including BsH, BsD, BsE, and BsW, the Folin–Ciocalteu colorimetric assay was carried out using gallic acid as a reference standard [55]. Shortly, 0.01 mL of each sample, at a concentration of 1 mg/mL, in methanol, was individually pipetted into 1.5 mL tube, followed by adding 0.20 mL of 10% Folin–Ciocalteu reagent and 1 mL of 10% sodium carbonate. After resting at RT for 20 min, each mixture was aliquoted into a 96-well plate, and we measured the absorbance at 765 nm with a spectrophotometer (DTX880 Multimode Detector, Beckman Coulter^®^, Wien, Austria). The results were done in triplicate and expressed as the mg of gallic acid equivalent/mg extract (mg GAE/mg extract).

### 3.5. Total Flavonoid Contents (TFC)

The total flavonoid contents of crude extract and four partitioned fractions were detected using quercetin as a reference standard by means of aluminum chloride colorimetric assay [56]. Briefly, 0.6 mL of each sample, at a concentration of 1 mg/mL, in methanol, was separately added into 5 mL tube, containing 1.25 mL of distilled water and 0.75 mL of 5% NaNO_2_. Then, the solution was incubated at RT for 5 min, followed by adding 0.15 mL of 10% AlCl_3_. Five minutes later, each reaction was thoroughly mixed with 0.5 mL of 1 M NaOH, and we measured the absorbance at 510 nm by a spectrophotometer (DTX880 Multimode Detector, Beckman Coulter^®^, Wien, Austria). The results were performed in triplicate and expressed as the mg of quercetin equivalent/mg extract (mg QE/mg extract).

### 3.6. Liquid Chromatography with Tandem Mass Spectrometry (LC-MS/MS) Analysis

Phenolic and flavonoid compounds in BsE fraction were identified by liquid chromatograph–quadrupole time-of-flight mass spectrometer (LC-QTOF MS, Agilent Technology, Santa Clara, CA, USA). The chromatographic separation was run on a Zorbax Eclipse Plus C18 column (150 × 2.1 mm, particle size 1.8 µm) at 25 °C. The injection volume was 2 µL, and the flow rate was kept at 0.2 mL/min. Elution was carried out by using two mobile phases: eluent A (0.1% formic acid in water) and eluent B (acetonitrile). The gradient program was as follows: 5% B (3 min), 23% B (22 min), 35% B (10 min), and 5% B (10 min).

MS was operated with Dual AJS source, and the instrument parameters were as follows: gas temperature, 325 °C; gas flow, 13 L/min; nebulizer, 35 psi; sheath gas temperature, 275 °C; sheath gas flow, 12 L/min; VCap, 4000 V; nozzle, 2000 V; fragmentor, 175 V; skimmer, 65 V; and octopole RF peak, 750. The scan range of the ion trap was 100–1500 m/z for MS and 50–1500 m/z for MS/MS. The mass spectra were recorded in both negative and positive ion modes. The reference mass ions purine at 112.9856 (for negative mode) and 121.0508 (for positive mode), as well as the HP921 at 1033.9881 (for negative mode) and 922.0098 (for negative mode), were used for continuously correcting any mass drift. All data were analyzed by LC/MS Data Acquisition and Qualitative Analysis Workflows software.

### 3.7. α-Glucosidase Inhibitory Assay

The inhibitory effects against α-glucosidase enzyme of crude extract, partitioned fractions, and all isolated pure compounds were examined according to previous assay [57]. In brief, 50 µL of each sample in 50 mM phosphate buffer, pH 6.9 (at various concentrations), was pre-mixed with 50 µL of 0.57 U/mL α-glucosidase enzyme for 10 min at 37 °C. Then, 50 µL of 5 mM pNPG substrate was added into the mixture and further incubated for 30 min, after that stopping the reaction by 50 µL of 1 M sodium carbonate solution (Na_2_CO_3_). A microplate reader (DTX880 Multimode Detector, Beckman Coulter^®^, Wien, Austria) immediately measured the absorbance at 405 nm. The following equation calculated the percentage of inhibition: [(A_control_ − A_sample_)/A_control_] × 100, where A_control_ and A_sample_ were the absorbance of reaction without and with the sample, respectively. Acarbose served as a positive control. The results were reported as a concentration of sample that could inhibit enzyme activity by 50% (IC_50_).

### 3.8. Dipeptidyl Peptidase-IV (DPP-IV) Inhibitory Assay

The DPP-IV inhibition activities of crude extract, partitioned fractions, and isolated pure compounds were investigated, following the procedure of Van et al. (2009) and Al-Masri et al. (2009), with some modifications [58]. Concisely, 40 µL of each sample diluted in 50 mM Tris-HCl buffer, pH 8.0 (at final concentration 50 ug/mL), was pre-incubated with 30 µL of 0.05 U/mL of DPP-IV enzyme at 37 °C. After 10 min, 100 µL of 0.2 mM GP-p-NA substrate was added to the solution and further incubated for 30 min. Then, the reaction was terminated with 30 µL of 25% acetic acid, and the absorbance at 405 nm was suddenly measured by a microplate reader (DTX880 Multimode Detector, Beckman Coulter^®^, Wein, Austria). The percentage of inhibition was calculated, following the formula: [(A_control_ − A_sample_)/A_control_] × 100, where A_control_ and A_sample_ were the absorbances of reaction with and without sample, respectively. Diprotin A was employed as a positive control, and the results were presented as the IC_50_ value.

### 3.9. Cell Culture

Raw 264.7 cells (murine macrophages) were purchased from the American Type Culture Collection (Manassas, VA, USA). They were grown in RPMI 1640 medium, supplemented with 10% FBS and 1% penicillin-streptomycin at 37 °C, under a humidified atmosphere containing 5% CO_2_. The medium was changed every 2–3 days and subcultured once the cells reached 80–90% confluence. The 3T3-L1 pre-adipocytes (mouse embryonic fibroblast) were generously given by the Medical Science Research and Innovation Institute, Prince of Songkla University. They were cultured in DMEM, high glucose, supplemented with 10% FBS and 1% penicillin-streptomycin and maintained in the same condition as the RAW 264.7 cells. These cells should be subcultured once the cells were reached 70% confluence, with the medium changed every two days.

### 3.10. Cell Differentiation

The 3T3-L1 pre-adipocytes were induced to differentiate into adult adipocytes after 48 h of confluence in 48 wells plate. The cells were incubated with cultured media, comprising of 1 µg/mL insulin, 10 µM dexamethasone, and 0.5 mM IBMX. After 48 h of incubation, differentiate media were removed and replaced with cultured media containing 1 µg/mL insulin. Then, the maintained media were changed every two days for eight days.

### 3.11. Cell Viability Assay

The cytotoxicity of crude extract, partitioned fractions, and isolated compounds was assessed in 3T3-L1 adipocytes and raw264.7 macrophage cells, according to previous methods [59]. Raw264.7 cells were seeded into 96-well plates, at a density of 1 × 10^5^ cells/well, and incubated at 37 °C for 2 h. The 3T3-L1 pre-adipocytes were seeded at a density of 5 × 10^3^ cells/well and maintained in the same condition for 24 h. After this step, the adhered cells were treated overnight with various concentrations of samples. Then, the treated medium was changed and further incubated with the MTT solution, at a final concentration of 0.5 mg/mL for 3 h. After that, the MTT-containing medium was removed and replaced with 100 µL of DMSO to dissolve the formazan crystals. Cell viability was monitored by measuring the absorbance at 570 nm. Results were presented as the percentage of control.

### 3.12. Glucose Uptake Assay

The glucose uptake stimulatory effect of crude extract, partitioned fractions, and isolated compounds was determined by the previously described method [60]. Briefly, differentiated adipocytes were grown in 48-well plates, with cultured media containing 2% FBS, for 18 h. Then, the cells were washed and treated with various concentrations of samples in a complete low glucose medium. After 24 h of incubation, the treated medium of samples was individually transferred into 96-well plates and mixed with the reagent of the GO kit at 37 °C for 30 min. The reaction was terminated by 6M H_2_SO_4_, and the absorbance of remaining glucose at 540 nm was recorded by a microplate reader. The ability to enhance glucose uptake was measured by the amount of glucose in the media. The percentage of glucose uptake stimulation was calculated by following equation: %stimulation = [(A_control_ − A_sample_)/A_control_] × 100, where A_control_ and A_sample_ were the absorbances of reaction with and without sample, respectively.

### 3.13. Inhibition of Nitric Oxide (NO) Production Using Griess Renitricaction Assay

The investigation of the inhibitory activities against LPS induced nitric oxide (NO) production of crude extract, partitioned fractions, and isolated compounds by Griess assay, which was accomplished as described by Kaewdana et al. [61]. Briefly, Raw264.7 cells were seeded, at the density of 1 × 10^5^ cells/well, in 96-well plates. After 1–2 h culture, the medium was changed and co-treated with different concentrations of samples and 200 ng/mL LPS for 24 h. After that, the nitrite-containing medium was equally mixed with the Griess reagent (1% sulfanilamide in 5% phosphoric acid and 0.1% naphthyl ethylenediamine dihydrochloride (NED) in water) and measured by a microplate reader at 570 nm. Indomethacin (a nonsteroidal anti-inflammatory drug) was used as a positive control. Results were shown as the percentage of inhibition calculated, following the equation: [(A_control_ − A_sample_)/A_control_] × 100.

### 3.14. Statistics

All data were expressed as mean *±* standard deviation and done in triplicate. Any significant difference was determined by one-way analysis of variance (one-way ANOVA), at *p* < 0.05, unless stated differently, using the GraphPad Prism 9 statistical package (GraphPad Software Inc., La Jolla, CA, USA).

## 4. Conclusions

This is the first study to investigate the different anti-diabetic effects of crude extract, partitioned fractions, and isolated compounds of *B. strychnifolia* stem. Using a traditional method, the ethanolic crude extract was obtained and considered for further separating into different fractions, including BsH, BsD, BsE, and BsW. The screening of TPC and TFC revealed that the BsE fraction from ethyl acetate contained the highest amounts of phenolics and flavonoids and might exert anti-diabetic effects. BsE manifested potent inhibitory activities against α-glucosidase and DPP-IV enzymes, as well as the highest rate of glucose uptake stimulation. The LC-MS/MS analysis demonstrated numerous phenolics and flavonoids in BsE fractions, and four isolated compounds from BsE, namely resveratrol, epicatechin, quercetin, and gallic acid, were also identified in both LC-MS/MS and NMR analysis. Among isolated compounds, quercetin displayed the maximum capacities in inhibition of both α-glucosidase and DPP-IV enzymes, as well as the stimulation of glucose storage in 3T3-L1 adipocytes. Our results suggested that quercetin was a promising bioactive compound, contributing to the blood glucose-lowering properties of BsE extract and the *B. strychnifolia* stem. It could be used as a bioactive marker of BsE for use as an alternative treatment T2DM. Our finding supports the safety and efficacy of the traditional use of the *B. strychnifolia* stem.

## Figures and Tables

**Figure 1 molecules-27-02393-f001:**
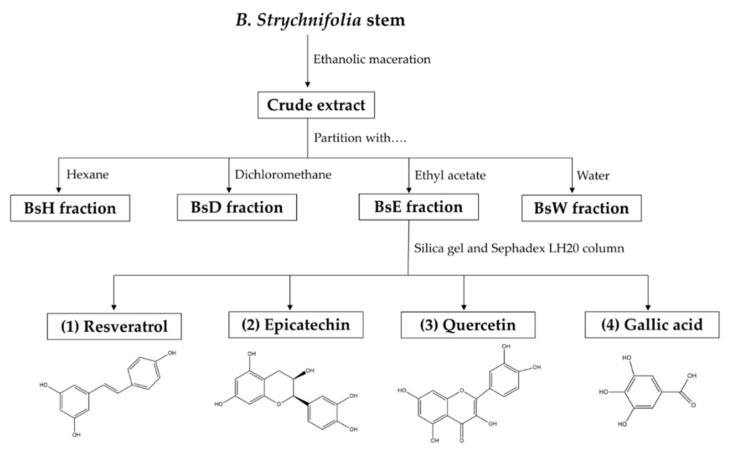
Schematic procedure of extraction, fractionation, and isolation of *B. strychnifolia* stem, as well as the chemical structures of compounds, which were isolated from the BsE fraction.

**Figure 2 molecules-27-02393-f002:**
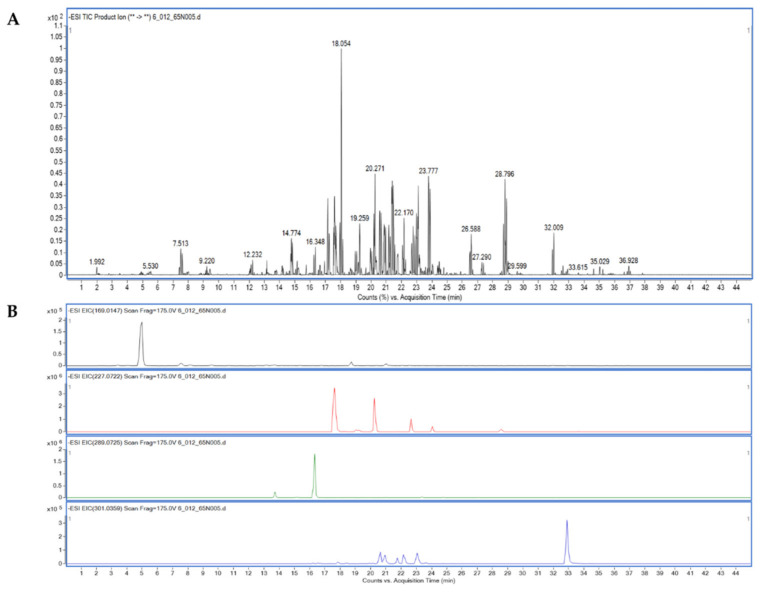
LC-MS/MS analysis. Total ion chromatogram (TIC) of phenolic compounds in BsE fraction (**A**) and extracted ion chromatogram (EIC) of four isolated compounds in Section 2.4, including gallic acid, resveratrol, epicatechin, and quercetin, respectively (**B**).

**Figure 3 molecules-27-02393-f003:**
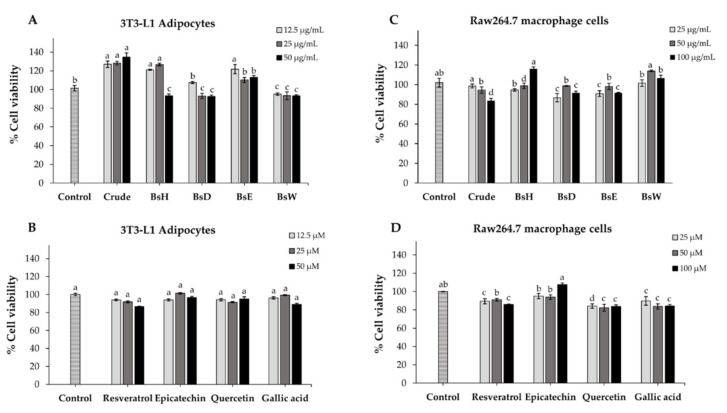
The percentage of cell viability of 3T3-L1 adipocytes (**A**,**B**) and raw264.7 macrophage cells (**C**,**D**) after treatment with different concentrations of crude extract, partitioned fractions, and isolated compounds. Values are expressed as the means ± SD (*n* = 3). Different letters (a–d) indicate statistical difference (*p* < 0.05).

**Figure 4 molecules-27-02393-f004:**
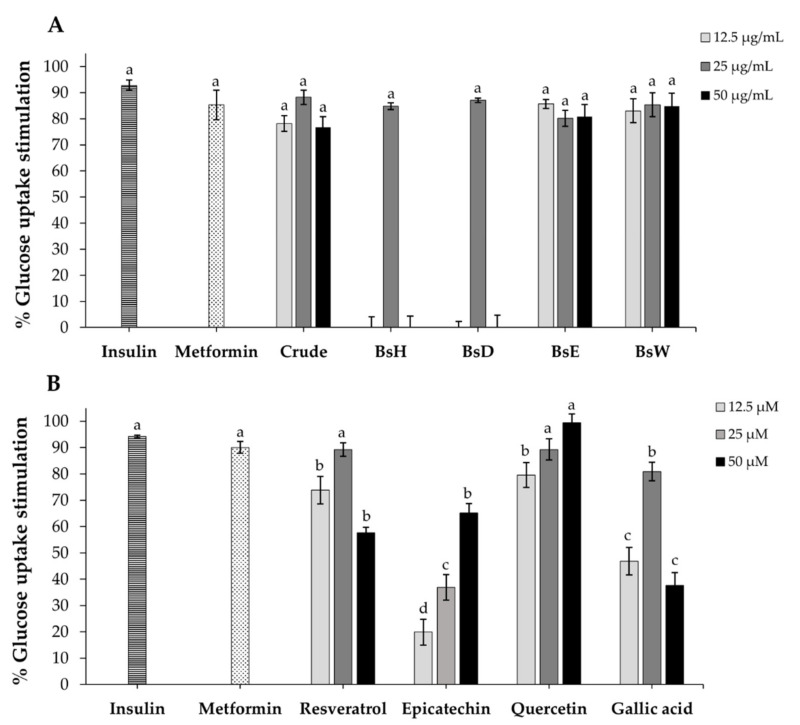
The percentage of glucose uptake stimulation in 3T3-L1 adipocytes after treatment with various concentrations of crude extract (**A**), partitioned fractions (**A**), and isolated compounds (**B**). Insulin (100 nM) and metformin (2 mM) were used as the positive control. Results are expressed as mean ± SD (*n* = 3). Different letters significantly indicate difference (*p* < 0.001).

**Figure 5 molecules-27-02393-f005:**
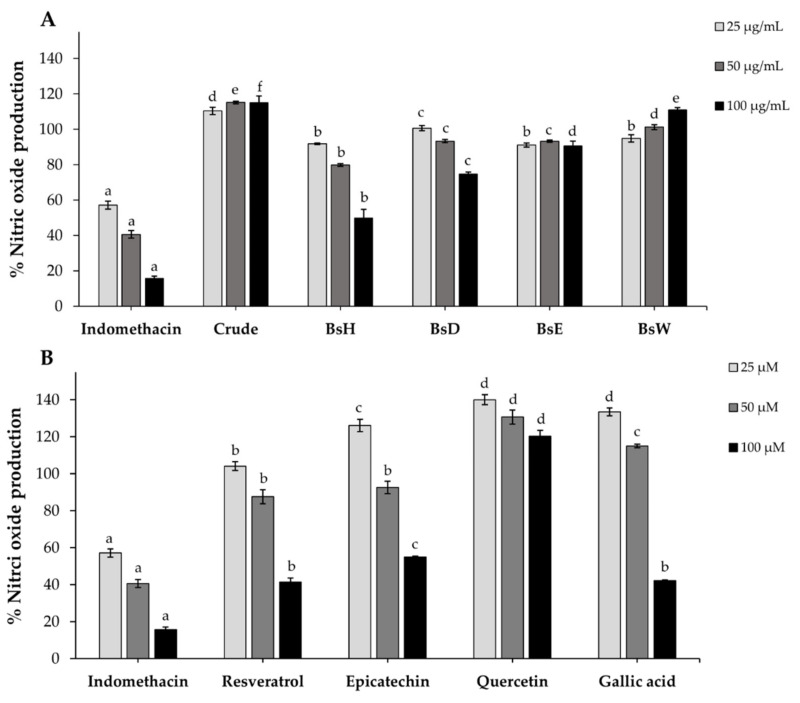
The percentage of nitric oxide production in LPS-induced Raw264.7 macrophage cells after 24 h of incubation with crude extract (**A**), partitioned fractions (**A**), and isolated compounds (**B**). Indomethacin was used as a positive control. Data are expressed as mean ± SD (*n* = 3). Different letters indicate a lack of significant difference (*p* < 0.001).

**Table 1 molecules-27-02393-t001:** Extraction yield, TPC, and TFC of crude extract and four partitioned fractions.

Extracts	% Yield Extraction	TPC mg GAE/g Extract	TFC mg QE/g Extract
Crude	10.30	562.33 ± 2.89	79.70 ± 0.72
BsH	0.40	12.33 ± 0.76	55.10 ± 0.00
BsD	8.15	210.67 ± 1.88	65.52 ± 0.72
BsE	5.90	620.67 ± 0.99 *	131.35 ± 2.50 *
BsW	60.00	392.33 ± 1.65	71.77 ± 0.72

GAE: gallic acid equivalent; QE: quercetin equivalent. * Indicates statistical difference, when compared with other fractions.

**Table 2 molecules-27-02393-t002:** Qualitative characterization of flavonoids and phenolics in BsE fraction.

Compounds	Formula	Retention Time (min)	Mode of Ionization (ESI^−^/ESI^+^)	Mass	*m/z*	Mass Error (ppm)
**Flavonoids**						
1. Epigallocatechin	C_15_H_14_O_7_	12.54	[M-H]^−^	306.07	305.07	0.02
2. Catechin	C_15_H_14_O_6_	13.72	[M-H]^−^	290.07	289.07	−0.05
3. Apuleirin	C_20_H_20_O_9_	15.00	[M-H]^−^	404.11	403.10	−0.40
4. Phloridzin	C_21_H_24_O_10_	15.11	[M-H]^−^	436.13	435.13	−0.05
5. Epicatechin	C_15_H_14_O_6_	16.35	[M-H]^−^	290.07	289.07	−0.57
6. Astilbin	C_21_H_22_O_11_	16.71	[M-H]^−^	450.11	449.11	−0.39
7. Rhapontin	C_21_H_24_O_9_	18.86	[M-H]^−^	420.14	419.13	−0.29
8. Quercetin3-(2-galloylglucoside)	C_28_H_24_O_16_	19.41	[M-H]^−^	616.11	615.10	0.18
9. Quercetin 3-galactoside	C_21_H_20_O_12_	20.68	[M-H]^−^	464.09	463.08	−1.01
10. Epicatechin Monogallate	C_22_H_18_O_10_	21.12	[M-H]^−^	442.09	441.08	−0.08
11. Taxifolin	C_15_H_12_O_7_	21.70	[M-H]^−^	304.05	303.05	−0.25
12. Kaempferol−7-o-glucoside	C_21_H_20_O_11_	22.21	[M-H]^−^	448.10	447.09	−1.07
13. Naringenin	C_15_H_12_O_5_	23.53	[M-H]^−^	272.07	271.06	−0.19
14. Alphitonin	C_15_H_12_O_7_	24.23	[M-H]^−^	304.06	303.05	−0.17
15. Apigenin 7-O-glucoside	C_21_H_20_O_10_	26.12	[M-H]^−^	432.11	431.11	−0.46
16. Phloretin	C_15_H_14_O_5_	28.84	[M-H]^−^	274.08	273.08	−0.58
17. Agehoustin C	C_22_H_24_O_10_	31.16	[M-H]^−^	448.13	447.13	0.31
18. Theasinensin C	C_30_H_26_O_14_	32.04	[M-H]^−^	610.13	609.12	3.00
19. Luteolin	C_15_H_10_O_6_	32.62	[M-H]^−^	286.04	285.04	−0.27
20. Quercetin	C_15_H_10_O_7_	32.92	[M-H]^−^	302.04	301.04	−0.12
21. Dichamanetin	C_29_H_24_O_6_	33.62	[M+HCOO]^−^	468.16	513.15	0.28
22. Viniferal	C_35_H_26_O_8_	35.07	[M+HCOO]^−^	574.17	633.18	−2.57
23. Cnidilin	C_17_H_16_O_5_	35.65	[M-H]^−^	300.10	299.09	0.09
24. Apigenin	C_15_H_10_O_5_	36.89	[M-H]^−^	270.05	269.04	0.14
25. Kaempferol	C_15_H_10_O_6_	37.90	[M-H]^−^	286.05	285.04	−0.17
**Phenolics**						
26. Gallic acid	C_7_H_6_O_5_	4.92	[M-H]^−^	170.22	169.01	−0.20
27. Caffeic acid	C_9_H_8_O_4_	10.58	[M-H]^−^	180.04	179.03	0.06
28. Gentisic acid	C_7_H_6_O_4_	13.19	[M-H]^−^	154.02	153.19	0.02
29. Kelampayoside A	C_20_H_30_O_13_	13.82	[M+HCOO]^−^	478.19	523.17	−0.08
30. Pyrocatechol	C_23_H_34_O_14_	15.41	[M-H]^−^	110.03	109.03	−0.04
31. Irisxanthone	C_20_H_20_O_11_	22.63	[M-H]^−^	436.10	435.09	−0.28
32. Resveratrol	C_14_H_12_O_3_	22.70	[M-H]^−^	228.07	227.07	−0.53
33. Isosyringinoside	C_23_H_34_O_14_	23.43	[M-H]^−^	534.19	533.19	0.56

**Table 3 molecules-27-02393-t003:** Inhibitory activities of the crude extract, partitioned fractions, and isolated compounds.

Samples	IC_50_	
	α-Glucosidase (µg/mL)	DPP-IV (µg/mL)
Crude extract	2.37 ± 0.13 ^a^	>50
BsH	10.61 ± 0.93 ^a^	>50
BsD	10.09 ± 0.75 ^a^	>50
BsE	1.51 ± 0.01 ^a^	2.62 ± 0.03 ^a^
BsW	2.42 ± 0.10 ^a^	3.20 ± 0.02 ^a^
Resveratrol	1.41 ± 0.02 ^a^ (8.16 µM)	3.22 ± 0.02 ^a^ (14.11 µM)
Epicatechin	>7.26 (25 µM)	>14.52 (50 µM)
Quercetin	1.89 ± 0.11^a^ (6.26 µM)	2.49 ± 0.01^a^ (8.25 µM)
Gallic acid	>4.25 (25 µM)	>8.50 (50 µM)
Acarbose	329.48 ± 6.91 ^b^ (509.60 µM)	-
Diprotin A	-	1.14 ± 0.05 ^a^ (3.26 µM)

Different letters (a,b) indicate statistical difference (*p* < 0.05).

## Data Availability

Not applicable.

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
