# Peer review of "Flavonoids and Phenols, the Potential Anti-Diabetic Compounds from *Bauhinia strychnifolia* Craib. Stem."

_molecules, 2022, doi:10.3390/molecules27082393_

Round 1

Reviewer 1 Report

The manuscript describes flavonoid and phenolics from Bauhinia strychnifolia stem. However, Identification of almost compounds of Table 2 is mistake except for resveratrol, epicatechin, quercetin and gallic acid which were identified by NMR. For example, in Table 2, formula and molecular weight of epicatechin is C15H14O6 and 290.27, those of catechin are C15H14O6 and 290.27, and those of naringenin are C15H12O5 and 272.26 and so on. So, I considered the manuscript as reject.

I considered the manuscript as reject.

Author Response

Reviewer 1

Comment 1

Identification of almost compounds of Table 2 is mistake except for resveratrol, epicatechin, quercetin and gallic acid which were identified by NMR. For example, in Table 2, formula and molecular weight of epicatechin is C15H14O6 and 290.27, those of catechin are C15H14O6 and 290.27, and those of naringenin are C15H12O5 and 272.26 and so on

Response #1

We apologize for the mistake. We have double-checked and improved all the data in Table 2.

Reviewer 2 Report

  1. traditional Thai medical plant for detoxification, was investigated for anti-diabetic. Sentence is incomplete.
  2. Therefore, BsE was further isolated its bioactive compounds using silica gel and Sephadex LH20 column. Rewrite this.
  3. The abstract needs to be updated as there are many flaws.
  4. Introduction section can be updated. Adding few relevant recent citations would be helpful.

https://doi.org/10.1016/j.dsx.2022.102441

https://doi.org/10.2174/1389201020666190731122806

https://doi.org/10.1021/acsomega.1c00631

  1. Previous studies demonstrated that most dietary flavonoids provided various medical activities, including anti-diabetic agents. A citation should be provided for the same.
  2. Table 2, Naringrnin. Is it naringenin of Naringrnin?
  3. In Table 2, under the heading Isolated phenol compounds, only gallic acid and resveratrol are mentioned.
  4. The authors did not include any in silico experiment. Once they were sure of the four isolated compounds, they could have performed in silico analysis of these four compounds.
  5. The manuscript has many grammatical and irrelevant use of abbreviations that need to be addressed.
  6. As the focus is on four isolated compounds from the extract, a little details about these should be added in the Introduction section.
  • Therapeutic Potential of Polyphenols in Alzheimer’s Therapy: Broad-Spectrum and Minimal Side Effects as Key Aspects.

Author Response

Reviewer 2

Comment 1

Traditional Thai medical plant for detoxification, was investigated for anti-diabetic. Sentence is incomplete.

Response #1

Thank you for your comment. We have rewritten the sentence for greater clarity. See below.

Original version

In this study, Bauhinia strychnifolia Craib. stem, a traditional Thai medical plant for detoxification, was investigated for anti-diabetic.

Revised version (Page 1)

In this study, Bauhinia strychnifolia Craib. stem, a traditional Thai medicinal plant for detoxification, was extracted into 5 fractions, including crude extract, BsH, BsD, BsE, and BsW by ethanolic maceration and sequential partition with hexane, dichloromethane, ethyl acetate, and water, respectively.

Comment 2

Therefore, BsE was further isolated its bioactive compounds using silica gel and SephadexLH20 column. Rewrite this.

Response #2

Thank you for your comment. We have rewritten the sentence for greater clarity. See below.

Original version

Therefore, BsE was further isolated its bioactive compounds using silica gel and SephadexLH20 column.

Revised version (Page 1)

Furthermore, the four compounds isolated from the BsE fraction, namely resveratrol, epicatechin, quercetin, and gallic acid, were identified.

Comment 3

The abstract needs to be updated as there are many flaws.

Response #3

Thanks for your comment. We have double-checked and rewritten the abstract.

Original version

Bioactive compounds from medicinal plants are good alternative treatments for T2DM or sources of lead molecules for new drug discovery. In this study, Bauhinia strychnifolia Craib. stem, a traditional Thai medical plant for detoxification, was investigated for anti-diabetic. The crude ethanolic extract was screened for hypoglycemic effects and further separated into BsH, BsD, BsE, and BsW fractions, according to the polarity partition. BsE contained the highest amounts of phenolics and flavonoids, considered likely bioactive compounds in B. strychnifolia. Consistently, BsE exhibited the dominant anti-diabetic effects such as inhibiting α-glucosidase with IC50 1.51 + 0.01 µg/mL, blocking dipeptidyl peptidase IV (DPP-IV) with IC50 2.62 + 0.03 µg/mL, and enhancing glucose uptake on 3T3-L1 adipocytes. Therefore, BsE was further isolated its bioactive compounds using silica gel and Sephadex LH20 column. Resveratrol, epicatechin, quercetin, and gallic acid were identified and investigated for glucose-lowering effects. The results showed that quercetin demonstrated the maximum inhibitory activity against α-glucosidase (IC50 6.26 + 0.36 µM) and DPP-IV (IC50 8.25 µM). Moreover, quercetin and resveratrol could promote glucose uptake stimulation effect on 3T3-L1 adipocytes. Altogether, we concluded that quercetin was probably a principle bioactive compound of B. strychnifolia stem for anti-diabetic activity.

Revised version (Page 1)

Bioactive compounds from medicinal plants are good alternative treatments for T2DM. They are also sources of lead molecules that could lead to new drug discoveries. In this study, Bauhinia strychnifolia Craib. stem, a traditional Thai medicinal plant for detoxification, was extracted into 5 fractions, including crude extract, BsH, BsD, BsE, and BsW by ethanolic maceration and sequential partition with hexane, dichloromethane, ethyl acetate, and water, respectively. Among these fractions, BsE contained the highest amounts of phenolics (620.67 mg GAE/ g extract) and flavonoids (131.35 mg QE/ g extract). BsE exhibited the maximum inhibitory activity against α-glucosidase (IC50 1.51 + 0.01 µg/mL) and DPP-IV (IC502.62 + 0.03 µg/mL), as well as dominantly promoting glucose uptake on 3T3-L1 adipocytes. Furthermore, the four compounds isolated from the BsE fraction, namely resveratrol, epicatechin, quercetin, and gallic acid, were identified. Quercetin demonstrated the highest inhibitory capacity against α-glucosidase (IC50 6.26 + 0.36 µM) and DPP-IV (IC50 8.25 µM). In addition, quercetin prominently enhanced the glucose uptake stimulation effect on 3T3-L1 adipocytes. Altogether, we concluded that quercetin was probably the principal bioactive compound of B. strychnifolia stem for anti-diabetic, and the flavonoid-rich fraction may be sufficiently potent to be an alternative treatment for blood sugar control.

Comment 4

Introduction section can be updated. Adding few relevant recent citations would be helpful.

Response #4

Thank you for your comment. We have added new citations in line 55.

Comment 5

Previous studies demonstrated that most dietary flavonoids provided various medical activities, including anti-diabetic agents. A citation should be provided for the same.

Response #5

Thank you for your comment. We have added a citation in line 97.

Comment 6

Table 2, Naringrnin. Is it naringenin of Naringrnin?

Response #6

We apologize for the mistake. It is naringenin. We have revised all the results in Table 2.

Comment 7

In Table 2, under the heading Isolated phenol compounds, only gallic acid and resveratrol are mentioned.

Response #7

Thank you for your comment. We have added more phenolic compounds data in Table 2.

Comment 8

The authors did not include any in silico experiment. Once they were sure of the four isolated compounds, they could have performed in silico analysis of these four compounds.

Response #8

Thank you for your comment. In this study, we did not perform an in-silico analysis of these four compounds since the molecular docking analysis between pure quercetin and both enzymes (α-glucosidase and DPP-IV) has been reported. We thought that pure quercetin in the previous reports and our quercetin is the same compound and will have the same result. We discuss it in lines 234-238.

Revised version

Although no prior studies tested the DPP-IV inhibitory activity in extract or compounds from B. strychnifolia stem, it has been reported that synthetic resveratrol and commercial quercetin could inhibit DPP-IV enzyme, and based on molecular docking, both gave high docking scores, which indicates that they could bind to the active site of DPP-IV [46,47].

Comment 9

The manuscript has many grammatical and irrelevant use of abbreviations that need to be addressed.

Response #9

            Thank you for your comment. We have rewritten the sentences in the revised version, as shown the example below.

Original version

In lines 43-46: Nowadays, many anti-diabetic drugs have been developed and have multiple modes of action to control blood sugar levels, such as inhibiting metabolic enzymes, like α-amylase and α-glucosidase, blocking dipeptidyl peptidase-IV (DPP-IV) enzyme, and enhancing glucose uptake.

In lines 55-61: Formerly, its leaves, root, and stem have been traditionally used as food, health tea drink, tonic, and traditional Thai medicine (TTM) for immunization, nourishing, alleviating fever, reducing alcoholic toxification, allergy treatment, eliminating toxins, anti-inflammation, anti-cancer and anti-diarrheal [14-17]. Several studies also reported its biological and pharmacological activities which are consistent with TMM knowledge such as antipyretic, alleviating allergy, detoxification, antioxidants, antimicrobial, anti-cancer, anti-HIV, and anti-diabetic [14-19].

In lines 63-69: The results showed that ethanolic extract of B. strychnifolia stem could inhibit α-glucosidase in vitro and reduce the levels of glucose, triglycerides, total cholesterol in the circulation of the alloxan-diabetic rats [16,17]. In addition, there were two flavonoids, namely 3,5,7,3’,5’-pentahydroxy-flavanonol-3-O-a-L-rhamnopyranoside and 3,5,7-trihydroxychromone-3-O-a-L-rhamnopyranoside were firstly isolated from ethanolic extract of B. strychnifolia stem and exhibited good inhibitory effect against α-glucosidase

In lines 70-72: Herein, we determine the different anti-diabetic effects of B. strychnifolia stem and explore its potent bioactive compounds.

In lines 102-104: The preliminary screen for bioactive compounds, the TPC and TFC of the crude extract, and four partitioned fractions were evaluated using colorimetric assays.

In line 110-112: Since phenolics and flavonoids are considered as principal constituents responsible for various biological activities [22].

In lines 122-124: To roughly identify and characterize flavonoids in BsE fraction, LC-MS/MS analysis was carried out by using the Liquid Chromatograph-Quadrupole Time-of-Flight Mass Spectrometer (LC-QTOF/MS).

In lines 137-139: Furthermore, we found that compounds isolated from BsE fraction (section 2.4), include resveratrol, epicatechin, quercetin, and gallic acid, which is consistent with our hypothesis (Figure 1B, Table 2).

In lines 206-208: Thus, it indicated that quercetin and resveratrol might be more potent than acarbose to reduce postprandial hyperglycemia, and they lower concentrations.

Revised version (Pages 1-2)

In lines 43-46: Recently, many anti-diabetic drugs have been developed and consist of multiple modes of action to control blood sugar levels, such as inhibiting metabolic enzymes, like α-amylase and α-glucosidase, blocking the dipeptidyl peptidase-IV (DPP-IV) enzyme, and enhancing glucose uptake.

In lines 55-61: Formerly, the leaves, roots, and stem of B. strychnifolia have all been used in foods, health tea drinks, tonics, and traditional Thai medicines (TTM) for immunization, nourishment, alleviating fever, reducing alcoholic toxification, allergy treatment, eliminating toxins, anti-inflammation, and anti-cancer and anti-diarrheal effects [14-17]. Several studies have also reported its biological and pharmacological activities, which are consistent with TTM knowledge such as antipyretic, alleviating allergy, detoxification, antioxidants, antimicrobial, anti-cancer, anti-HIV, and anti-diabetic [14-19]. 

In lines 63-69: It was found that the ethanolic extract of the B. strychnifolia stem inhibited α-glucosidase in vitro and reduced glucose levels, triglycerides, and total cholesterol in the circulation of the alloxan-diabetic rats [18,19]. In addition, the two flavonoids, namely 3,5,7,3’,5’-pentahydroxy-flavanonol-3-O-a-L-rhamnopyranoside and 3,5,7-trihydroxychromone-3-O-a-L-rhamnopyranoside, that were isolated from the ethanolic extract of the B. strychnifolia stem also exhibited good inhibitory effect against α-glucosidase

In lines 70-72: Herein, we determined the different anti-diabetic effects of the B. strychnifolia stem and explored its potent bioactive compounds.

In lines 102-104: To preliminary screen for the bioactive compounds in each fraction, the TPC and TFC of crude extract, and four partitioned fractions, were evaluated using colorimetric assays.

In lines 110-112: Phenolics and flavonoids in medicinal plants and foods were principal constituents responsible for various anti-diabetic activities [22].

In lines 122-124: Liquid Chromatograph-Quadrupole Time-of-Flight Mass Spectrometer (LC-QTOF/MS) was used to identify flavonoid and phenolic components in BsE fraction.  Based on the retention time (RT) comparison, mass spectrometric obtained under both negative and positive electron spray ionization modes (ESI-/ESI+), mass error, and data identification score, 25 flavonoids and 8 phenolics were preliminarily identified. Flavonoids in BsE fraction have been reported as anti-diabetic agents, like quercetin, kaempferol, naringenin, apigenin, luteolin (Table 2) [26-38].

In lines 137-139: Furthermore, we found that the compounds isolated from the BsE fraction (section 2.4), which included resveratrol, epicatechin, quercetin, and gallic acid, are consistent with our hypothesis (Figure 1B, Table 2).

In lines 206-208: Thus, quercetin and resveratrol might be more potent than acarbose to reduce postprandial hyperglycemia.

Comment 10

As the focus is on four isolated compounds from the extract, a little detail about these should be added in the Introduction section.

Response #10

Thank you for your comment. We have added some details about these isolated compounds, which are phenolics and flavonoids, in lines 52-54, as shown below.

Revised version (Page 2)

            In lines 52-54: It contained high phenolic and flavonoid compounds, which are considered likely bioactive compounds for various biological activities, including anti-diabetic [12,13,17].